# Evaluation of Amiodarone Administration in Patients with New-Onset Atrial Fibrillation in Septic Shock

**DOI:** 10.3390/medicina60091436

**Published:** 2024-09-02

**Authors:** Andreea Oprea, Virginia Marina, Oana Roxana Ciobotaru, Cristina-Mihaela Popescu

**Affiliations:** 1“Sf. Ioan” Children’s Emergency Hospital, 800487 Galati, Romania; andreeapintilie4@gmail.com; 2Doctoral School of Faculty of Medicine and Pharmacy, “Dunărea de Jos” University, 47 Str. Domnească, 800201 Galati, Romania; 3Department of Clinical Medical, Faculty of Medicine and Pharmacy “Dunarea de Jos” University, 800008 Galati, Romania; roxana.ciobotaru@ugal.ro; 4Dental-Medicine Department, Faculty of Medicine and Pharmacy, “Dunărea de Jos” University, 800008 Galati, Romania; cristina.popescu@ugal.ro

**Keywords:** amiodarone, new-onset atrial fibrillation, calcium channel blockers

## Abstract

*Background and Objective*: New-onset atrial fibrillation (NOAF) is a common cardiac condition often observed in intensive care units. When amiodarone is used to treat this condition, either to maintain sinus rhythm after electrical cardioversion or to control heart rate, complications can arise when a systemic pathology is present. Systemic pathology can result in a decrease in cardiac output and blood pressure, making the management of NOAF and septic shock challenging. Limited international research exists on the coexistence of NOAF and septic shock, making it difficult to determine the optimal course of treatment. While amiodarone is not the primary choice of antiarrhythmic drug for patients in septic shock, it may be considered for those with underlying cardiac issues. This paper aims to investigate the safety of administering amiodarone to patients with septic shock and explore whether another antiarrhythmic drug may be more effective, especially considering the cardiac conditions that patients may have. *Materials and Methods*: To write this article, we searched electronic databases for studies where authors used amiodarone and other medications for heart rate control or sinus rhythm restoration. *Results*: The studies reviewed in this work have shown that for the patients with septic shock and NOAF along with a pre-existing cardiac condition like a dilated left atrium, the use of amiodarone may provide greater benefits compared to other antiarrhythmic drugs. For patients with NOAF and septic shock without underlying heart disease, the initial use of propafenone has been found to be advantageous. However, a challenge arises when deciding between rhythm or heart rate control using various drug classes. Unfortunately, there is limited literature available on this specific scenario. *Conclusions*: NOAF is a frequent and potentially life-threatening complication occurring in one out of seven patients with sepsis, and its incidence is rising among patients with septic shock.

## 1. Introduction

Maintaining the heart rate within physiological limits is a crucial aspect of managing septic shock in patients hospitalized in the intensive care unit. This focus on heart rate regulation may enhance medium- and long-term survival outcomes. Despite various studies exploring optimal treatments to keep heart rate within physiological limits for patients with septic shock, a consensus has not yet been reached. Traditional medications like beta-blockers and calcium channel blockers, commonly used to restore normal heart rhythm in patients with atrial fibrillation and septic shock, should be avoided due to their negative impacts on inotropism and blood pressure. An alternative medication, amiodarone, can effectively control heart rate with fewer side effects on inotropism and hemodynamics [1].

The immune system’s responses to infection manifest as sepsis and septic shock, potentially leading to multiple organ dysfunction [2]. The mortality rates for sepsis and septic shock are 20.8% in intensive care units and 24.9% in other departments. Among patients discharged with a septic shock diagnosis, 58.3% succumbed to the condition. Notably, patients hospitalized in the intensive care unit and patients who remained in the hospital for more than six days faced significantly higher mortality rates after discharge [3]. Following the onset of septic shock and an exaggerated immune response, a surge of catecholamines triggers pathological changes such as the following: sustained tachycardia, cardiac rigidity, diastolic phase shortening, decreased ejection fraction, and reduced vascular volume in advanced stages [2]. Sepsis can cause cardiac arrhythmias, with atrial fibrillation being the most common. It is estimated that approximately 50% of patients diagnosed with sepsis experience recurring episodes of atrial fibrillation within the first year.

During the first three days of hospitalization in the intensive care unit, septic patients are six times more likely to develop AF. Those with sepsis who developed atrial fibrillation had a three times higher risk of Ischemic Stroke compared to septic patients without atrial fibrillation [4,5]. The prevalence of sepsis associated with the development of atrial fibrillation ranges from 2% to 26%, with a prevalence of over 40% in patients with septic shock [5,6,7]. The higher prevalence of atrial fibrillation has been observed in septic patients who are older and have multiple cardiovascular comorbidities [8]. The exact mechanism by which sepsis and septic shock lead to the onset or unmasking of atrial fibrillation is not fully understood, but studies have shown an increase in C-reactive protein before the onset of atrial fibrillation, indicating a close relationship between inflammation, sepsis, and AF [1,5,6].

Amiodarone is a drug that can be used to control supraventricular arrhythmias by blocking of calcium channels and β-receptors, while also having a neuroprotective effect [2,9,10]. It is considered effective in targeting the pathologic mechanism of tachycardia induced by sepsis or septic shock, with a few reported cases of amiodarone causing blood pressure instability, unlike beta-blockers, which can cause hypotension [2,11].

Amiodarone is also used to restore sinus rhythm in septic patients with atrial fibrillation, with limited adverse effects on the cardiovascular system [2,12].

This narrative review aims to assess whether the administration of amiodarone is the safest way to control heart rate in septic shock patients, highlighting the latest data on restoring sinus rhythm in patients. Due to limited data on amiodarone dosing, administration methods, time to cardioversion, and arrhythmia recurrence rates, this article poses a challenge for future research in the field.

## 2. Relevant Sections

We conducted a comprehensive review of the current literature, focusing on original articles that discussed the administration of amiodarone in septic patients to restore sinus rhythm. Extensive searches were performed on the Google Scholar, PubMed, Science Direct, Scopus, Elsevier, Web of Science, and Cochrane databases to identify relevant manuscripts. Key words such as “amiodarone”, “sepsis”, “septic shock”, “sinus rhythm”, “cardioversion”, “antiarrhythmics”, and “atrial fibrillation” were used to screen results, which were then complied into electronic files. The search was limited to English language publications, resulting in over 30 relevant manuscripts on the topic of amiodarone administration in patients with septic shock. We selected articles focusing solely on chronic atrial fibrillation, and removed duplicate articles and incomplete dissertation papers.

We then performed a detailed review of titles and abstracts to determine their compatibility with our inclusion criteria. The selection criteria were based on reports and case studies involving patients with sepsis or septic shock and atrial fibrillation who were treated with amiodarone or another antiarrhythmic to restore sinus rhythm. We systematically applied the selection criteria to evaluate the studies based on their journal, publication date, case report quality, results, and conclusions. Non-English studies were eliminated.

After reviewing and evaluating 36 articles, we identified and included a subset of six papers directly relevant to our study (Table 1). These selected studies provided valuable insights into the restoration of sinus rhythm or heart rate control in patients with newly diagnosed septic shock and atrial fibrillation, highlighting optimal treatment and the challenges faced. These challenges are summarized in Figure 1.

### 2.1. Main Concerns Regarding the Restoration of Sinus Rhythm or Heart Rate Control in Patients with Septic Shock and Newly Diagnosed NOAF

#### 2.1.1. Prevalence of NOAF in Septic or Septic Shock Patients

Septic shock is a condition characterized by an exaggerated inflammatory response to an infectious agent, resulting in severe arterial hypotension, inadequate perfusion pressure in various organs, and severe organ injury [13,14,15,16]. In this pathology, afterload is reduced, leading to left ventricular systolic and diastolic dysfunction. This also causes an increase in catecholamine release and chronotropic dysregulation, all of which can lead to arrhythmias. The most common arrhythmia occurring in this context is newly diagnosed atrial fibrillation, accounting for over 70% of supraventricular arrhythmias in septic shock. Typically, arrhythmias onset within the first 72 h of septic shock [6,17,18]. According to a study conducted by Peter M C Klein Klouwenberg et al., on 1782 patients hospitalized in the intensive care unit from 2011 to 2013, the risk of NOAF is 10% for sepsis patients, 22% for severe sepsis patients, and 40% for septic shock patients. New-onset atrial tachyarrhythmia (NOAF) is associated with increased ICU hospitalization time and a higher risk of death [6].

Another study titled “New onset atrial fibrillation in patients with sepsis”, conducted through a systematic literature search in MedLine, concluded that over 20% of septic shock patients develop NOAF, about 8% of all sepsis patients receiving this diagnosis [19]. A meta-analysis of 225.841 patients from 13 individual studies showed that NOAF in the context of sepsis is linked to an increased risk of stroke and higher in-hospital mortality and post-discharge mortality. The incidence of NOAF was 1.9% for patients with mild sepsis and rose to 46% in septic shock patients [20]. The average incidence of NOAF in sepsis was 20.6%, with a rate of 31.6% in prospective studies and 14.7% in retrospective studies. One of the seventeen studies included in a review published in 2021 reported an in-hospitalization stroke incidence of 6% in patients with NOAF, compared to only 0.69% in patients without AF. Of the seventeen studies, seven provided data on follow-up after discharge. In three of the studies, NOAF was associated with a high degree of mortality after discharge, specifically 34% at 28 days, 21% at 1 year, and 3% at 5 years [21].

The prevalence of NOAF in septic shock appears to be higher in males, a conclusion supported by the study “Use of Amiodarone in Management of Atrial Tachyarrhythmia in Septic Shock”, conducted on a sample of 65 patients. The study showed that 56% of patients with NOAF and septic shock were male and 68% of patients with this diagnosis had preserved left ventricular ejection fraction (FEV1 > 40%) [22].

The study by Liu, W.C. et al. shows that the incidence of NOAF in septic patients is 35.1%, higher than the results of other previous studies which ranged from 10% to 46% [23,24]. Another large study conducted on 39.693 patients showed that the mean age for NOAF in sepsis is 77 years (±11), and the majority (51%) were male and white race (76%) [25].

The prevalence of new-onset atrial fibrillation in sepsis ranges between 10% and 41%. It occurs more frequently in male patients with preserved left ventricular ejection fraction, Caucasians, and those with an average age of 77 years. In patients with septic shock, the prevalence of NOAF is even higher, reaching up to 46%.

#### 2.1.2. Most Commonly Used Mode of Administration and Dose of Amiodarone

The retrospective post-hoc study by Kevin B. et al. showed that the dose of amiodarone administered in the ICU was closely related to the risk of needing to continue this drug therapy after discharge (*p* < 0.01). Of the 118 patients in the study, 105 patients were discharged, and only 29% of these patients (*n* = 30) did not need to continue amiodarone drug therapy at home. It was demonstrated that the higher the dose of amiodarone administered in the ICU (6.550 g vs. 2.295 g), the higher the risk of needing to continue antiarrhythmic therapy at home [26].

#### 2.1.3. Amiodarone Studies

Another study, in a sample of 209 patients who were treated with amiodarone or propafenone, used intravenous medication as the mode of administration. Both groups of patients were given either an amiodarone intravenous bolus of 300 mg, followed by 600–1800 mg/24 h, or a propafenone intravenous bolus of 70 mg, followed by 400–840 mg/24 h [27]. A study published in 2024 and conducted on 170 patients also used the same mode and drug doses for both amiodarone and propafenone [28]. For oral administration, a study in a sample of 200 patients demonstrated high efficacy in converting NOAF to sinus rhythm for propafenone (85% of the 100 patients went into sinus rhythm) (*p* = 0.699), compared to those treated with intravenous amiodarone (83%) [29]. Another study of 239 patients showed that administration of ≥2700 mg of intravenous amiodarone was independently associated with a longer duration of hospitalization in the ICU [1].

#### 2.1.4. Hemodynamic Changes within the First 72 Hours of Amiodarone Administration

Supraventricular arrhythmias that suddenly occur in septic shock can lead to hemodynamic issues [28]. According to the “Amiodarone for Atrial Fibrillation in Septic Shock” study involving 44 patients, there were no significant differences in hemodynamic changes within the first 72 h after intravenous amiodarone administration between patients with septic shock (*n* = 12) and those without septic shock (*n* = 32) experiencing NOAF [30]. A retrospective study from 2019 suggests that propafenone may be more effective in restoring sinus rhythm without negatively impacting hemodynamics [31].

#### 2.1.5. Time to Conversion to Sinus Rhythm

The success rate of cardioversion in patients with septic shock ranges from 70% to 87% [17,18]. A study by Balik, M. et al. involving 209 patients, with 100 receiving amiodarone and 100 receiving propafenone, found that sinus rhythm was restored in an average of 3.7 h for propafenone (95% of patients) and 7.3 h for amiodarone (95% of patients). After 24 h, 77 (72.8%) and 71 patients, respectively, were in sinus rhythm. Patients with a dilated left atrium treated with amiodarone achieved better rhythm control (6.4 h) compared to those treated with propafenone (18 h, *p* = 0.05) [27].

Another study published in 2024 showed that patients with a dilated left atrium (*n* = 37 out of 170 patients) had faster rhythm control with amiodarone, while patients with a non-dilated left atrium (133 patients) achieved faster cardioversion with propafenone [29].

A comparative study from 2019 demonstrated that propafenone was superior in achieving sinus rhythm within the first 24 h, with 89% of patients on propafenone converting to sinus rhythm, compared to 74% on amiodarone [31].

In terms of modes of administration, a study of 200 patients found that intravenous amiodarone took longer to convert NOAF to sinus rhythm compared to oral propafenone [29].

#### 2.1.6. Mortality at 28 Days

The mortality rate in the ICU for patients with septic shock and NOAF who respond to antiarrhythmic medication is 33.5%, increasing to 43.6% at 28 days after NOAF onset. There are no significant differences in mortality compared to patients with septic shock and chronic atrial fibrillation, with ICU mortality at 38.2%, rising to 41.4% at 28 days [17,18]. A univariate analysis from a 2017 study showed that long-term mortality was similar in patients treated with propafenone or metoprolol and was lower than in those treated with amiodarone [17].

A recently published study of 170 patients showed that patients with end-systolic left atrial volume (LAVI) <40 mL/m^2^ (133 patients) who were given propafenone for supraventricular arrhythmia had lower one-year mortality rates (*p* = 0.014). However, patients with LAVI ≥40 mL/m^2^ treated with propafenone had less favorable one-month mortality outcomes compared to those treated with amiodarone (*p* = 0.045) [28].

Mortality in the intensive care unit among patients with sepsis or septic shock associated with NOAF is high, and it increases if sinus rhythm restoration fails. A study conducted on 503 patients (263 without NOAF and 240 with NOAF) hospitalized in the ICU ward between 2011 and 2014, revealed that the in-hospital mortality rate for patients with NOAF in whom sinus rhythm could not be restored was 61.3%, significantly higher compared to the group in whom drug cardioversion occurred (26.1%) or those without NOAF (17.5%). Multivariate logistic regression analysis showed that failure to restore sinus rhythm is independently associated with an increase in in-hospital mortality in septic patients with NOAF [23]. The risk of mortality for septic patients with NOAF in whom drug cardioversion did not occur was 2.22 times higher than in those in whom sinus rhythm was restored. The mortality risk for the group with NOAF and restored sinus rhythm and the group without NOAF was statistically similar (*p* = 0.22) [23].

#### 2.1.7. Recurrent NOAF in Septic Shock

Recurrent NOAF in septic shock is a common issue, with approximately 30–35% of patients developing chronic AF despite treatment [6,18,19]. A study by Martin B et al. on a sample of 209 patients treated with either amiodarone or propafenone showed that arrhythmia recurred in 76% of patients treated with amiodarone and in 52% of patients treated with propafenone (*p* < 0.001) [27].

Studies have identified certain echocardiographic parameters that may be prognostic in maintaining sinus rhythm after drug cardioversion [32]. A total of 37 patients with a dilated right atrium, in a study on a sample of 170 patients, demonstrated similar relapse rates for treatment with amiodarone or propafenone (*p* = 0.5) [28].

#### 2.1.8. Amiodarone or Another Antiarrhythmic

There is insufficient information on managing septic shock patients with vasopressor requirements and rapid rhythm/rapid ventricular rate NOAF [22]. The optimal regimen for NOAF in septic shock remains uncertain, with amiodarone being commonly used in such cases [26].

A study by Jaffer, F. et al. on a sample of 298 individuals, of whom only 65 met the inclusion criteria (patients over 18 years old with septic shock and rapidly developed NOAF within the first 48 h of admission), showed that there were no significant differences in the mortality rate at ICU discharge for patients treated with amiodarone, beta-blockers, or calcium channel blockers. Patients were divided into four categories according to the medication administered: 49% of the 65 patients were treated with amiodarone, 15% were treated with calcium channel blockers, 24% were in the control group, and 12% were treated with beta-blockers. A total of 56% of the patients studied were male, and 68% of them had a left ventricular ejection fraction (LVEF) greater than 40%. The results of the Simplified Acute Physiology Score II (SAPSII) were similar for all four groups. In terms of the number of days of persistent NOAF, total days of mechanical ventilation, duration of hospitalization, duration of ICU admission, and ischemic events in the ICU, all were similar in the four groups studied. One difference that could be observed was the Model for End-Stage Liver Disease (MELD) score adjusted for age and sex, which was significantly higher (20.85 ± 8.70) for the amiodarone group (*p* = 0.03) compared to the control group (15.40), or those treated with beta-blockers (12.88) or calcium channel blockers (17.10). Thus, the study concluded that for patients with hepatic dysfunction, other antiarrhythmic agents besides amiodarone should be considered, in an effort to limit its use in these patients [22].

Another study conducted on 39.693 patients concluded that the most commonly used drug for NOAF in sepsis was calcium channel blockers, at 36% (*n* = 14.202 patients). The study compared groups of patients given calcium channel blockers, beta-blockers, amiodarone, or Digoxin. The second most commonly used drug was beta-blockers in 28% of patients (*n* = 11.290), followed by Digoxin in 7.937 patients (20%), and amiodarone in 16% of patients (*n* = 6264). In terms of mortality, the study demonstrated that although calcium channel blockers were the most commonly used treatment, beta-blockers were associated with lower mortality compared to amiodarone (*n* = 5.378), Digoxin (*n* = 13.991), and calcium channel blockers (*n* = 18.720) [25] Thus, this study provides a rationale for clinical trials comparing the efficacy of rhythm- or frequency-targeted therapy in NOAF sepsis. In sepsis, the best results have been obtained with beta-blockers, but in septic shock, the results may be very different. Due to the fact that in these patients, vasopressor support (norepinephrine and/or dobutamine) is necessary, beta-blockers may cancel their effect [33].

In the study by Balik, M. et al., it was observed that propafenone, compared to amiodarone, does not provide better rhythm control at 24 h after the onset of NOAF, but it achieves faster cardioversion (3.7 h) with fewer recurrences (52% of patients had a recurrence of arrhythmia), especially in patients with a non-dilated left atrium. In patients with a dilated left atrium, better rhythm control until cardioversion (6.4 h) was obtained with intravenous administration of amiodarone [27].

Regarding the safety of amiodarone administration, it may have important side effects such as diastolic dysfunction; decreased contractility and decreased cardiac output (negative inotropism) hepatic dysfunction, occurring in 1–2% of patients with intravenous administration; corneal micro-deposition; pulmonary fibrosis that can occur in 5–10% of cases and can slowly progress to interstitial pneumonia with diffuse bilateral infiltrates; neuropathy; and skin discoloration [18,34,35,36,37].

If we consider the mode of administration of antiarrhythmic agents, a study published in 2022 on a sample of 200 patients demonstrated that orally administered propafenone (*n* = 100) achieved conversion to sinus rhythm faster than intravenously administered amiodarone (*n* = 100). For the oral propafenone group, NOAF conversion to sinus rhythm was 85%, and for those on intravenous amiodarone it was 83% [29].

## 3. Results

The results of the present study, which is based on a subset of six studies on the efficacy of amiodarone or other medications in patients with septic shock and NOAF, are summarized in Table 1.

**Table 1 medicina-60-01436-t001:** Efficacy of medication in patients with septic shock and NOAF.

Article Type and Authors	Number of Patients or Studies Used	Treatment Used in the Control Groups	Use of Amiodarone in the Study Group	Objectives	Results
Retrospective chart review. Jaffer, F. et al. [22]	65 patients (4 groups)	Beta-blocker Calcium channel blocker	iv.	Mortality at hospital discharge. Number of days with persistent NOAF. Number of days on mechanical ventilation. Duration of intensive care and hospitalization. Occurrence of adverse reactions.	The SAPSII score was similar for all four groups. There were no significant differences in mortality upon discharge, the number of days of NOAF, the number of days of M.V., the number of days in intensive care, or ischemic events. The MELD score adjusted for sex and age at the end of hospitalization was significantly higher (*p* = 0.003) for the amiodarone group (20.85) compared to the control group (15.40), the beta-blocker group (12.88), and the calcium channel blocker group (17.10).
Double-blind, prospective controlled study in two centers. Balik, M. et al. [27]	209 patients (2 groups)	Propafenone	iv.	The proportion of patients in whom sinus rhythm was restored within 24 h after the start of drug infusion. Time taken to achieve sinus rhythm. Proportion of patients experiencing recurrence of arrhythmia.	Within 24 h of starting drug infusion, 72.8% of patients on propafenone and 67% of patients on amiodarone were in sinus rhythm (*p* = 0.4). The mean time to sinus rhythm was 3.7 h for propafenone and 7.3 h for amiodarone (*p* = 0.02). The arrhythmia recurrence rate was 76% (*n* = 80) for patients on amiodarone and 52% for patients on propafenone (*p* < 0.001). Patients with a dilated left atrium had better rhythm control until cardioversion with amiodarone (6.4 h) than those treated with propafenone (18 h) (*p* = 0.05). Propafenone provides faster cardioversion with fewer arrhythmia recurrences, but the 24-h control is not better compared to amiodarone, especially in patients with a dilated left atrium. Patients with a non-dilated left atrium (*n* = 133) went into sinus rhythm faster (*p* = 0.009) and with fewer recurrences (*p* = 0.001) following administration of propafenone (70 mg bolus, then 400–840 mg i.v./24 h) compared to those treated with amiodarone (300 mg bolus i.v., then 600–1800 mg/24 h). In patients with LAVI (end-systolic left atrial volume) <40 mL/m^2^ and with propafenone, one-year mortality was lower compared to those treated with amiodarone (*p* = 0.014).
Randomized controlled trial. Waldauf, P. et al. [28]	170 patients (2 groups)	Propafenone	iv.	Comparison between the speed of cardioversion and the rate of recurrence in patients with a dilated or non-dilated left atrium, as well as one-month and one-year mortality.	Patients with a non-dilated left atrium (*n* = 133) who received propafenone (70 mg bolus followed by 400–840 mg i.v./24 h) entered sinus rhythm quicker (*p* = 0.009) and experienced fewer recurrences (*p* = 0.001) compared to those treated with amiodarone (300 mg bolus i.v. followed by 600–1800 mg/24 h). In patients with LAVI (end-systolic left atrial volume) <40 mL/m^2^ who were treated with propafenone, one-year mortality was lower than in those treated with amiodarone (*p* = 0.014). Patients with a dilated left atrium (*n* = 37) who were treated with amiodarone achieved sinus rhythm sooner than those treated with propafenone. The relapse rate was similar in both groups (*p* = 0.5). Among patients with LAVI ≥40 mL/m^2^ treated with propafenone, 1-month mortality was higher compared to the amiodarone-treated group (*p* = 0.045). However, there was no statistically significant difference in 1-year mortality (*p* = 0.138).
Randomized controlled trial. Taha, H.S. et al. [29]	200 patients (2 groups)	Propafenone	iv.	Comparison of the speed of conversion of NOAF to sinus rhythm using two medications (oral propafenone and intravenous amiodarone). Comparison of the efficacy of two antiarrhythmic drugs.	The rate of NOAF to sinus rhythm conversion was 83% in the intravenous amiodarone group and 85% in the oral propafenone group (*p* = 0.699). The time to onset of sinus rhythm after amiodarone was 9.07 h (±5.04), and 3.9 h (±1.54) after propafenone (*p* = 0.001). Significantly larger left atrial diameters and higher CRP (C-reactive protein) levels were recorded in both groups with failed cardioversion.
Propensity-matched cohort study. Walke, A.J. et al. [25]	39.693 patients (4 groups)	Beta-blocker Calcium channel blocker Digoxin	iv.	Comparison of in-hospital mortality for the 4 groups in which drug administration was intravenous. Which drug was most commonly used for treating NOAF in sepsis?	Calcium channel blockers were the most commonly used medication for NOAF in sepsis (36%, *n* = 14.202), followed by beta-blockers in 11.290 patients (28%), Digoxin in 20% of patients (*n* = 7.937), and amiodarone in only 16% of cases (*n* = 6.264). The highest in-hospital mortality rate was among patients treated with calcium channel blockers (*n* = 18.720), followed by those treated with Digoxin (*n* = 13.994) and amiodarone (*n* = 5.378). Patients treated with beta-blockers had the lowest in-hospital mortality rate. Calcium channel blockers were the most commonly used medication for NOAF in sepsis, with 36% of patients (*n* = 14.202) receiving this treatment. Beta-blockers were used in 11.290 patients (28%), Digoxin in 20% of patients (*n* = 7.937), and amiodarone in only 16% of cases (*n* = 6.264). The highest in-hospital mortality rate was among patients treated with calcium channel blockers (*n* = 18.720), followed by those treated with Digoxin (*n* = 13.994) and amiodarone (*n* = 5.378). Patients treated with beta-blockers had the lowest in-hospital mortality rate.
Retrospective Study. Balik M. et al. [17]	234 patients (3 groups)	Propafenone Metoprolol	iv.	Comparing the cardioversion rates in 3 groups, and the rates of intensive care hospitalization and mortality at 28 days and 12 months.	99.1% of patients in the study were on mechanical ventilation. 69.7% developed NOAF. The cardioversion rate was 74% in group I (amiodarone), 89% in group II (propafenone), and 92% in group III (metoprolol). The period of hospitalization in the ICU and the 28-day mortality rate were not significantly different in the 3 groups. Using multivariate analysis, the 12-month mortality was higher in group I than in group II (*p* = 0.03). Patients with unsuccessful cardioversion did not show a significantly higher mortality rate in the ICU at 28 days or at one year compared to those with successful cardioversion or chronic AIF.

## 4. Discussions

The prevalence of NOAF is varied among sepsis patient groups, with higher rates seen in septic shock patients [25]. Echocardiography can help guide the management of NOAF in septic shock by monitoring the recovery of atrial function after cardioversion. Two-dimensional LA and Doppler functional parameters are important in assessing the risk of arrhythmia recurrence. Specifically, a left atrial ejection fraction (LA-EF) of >44% and a wave velocity-time-integral (Avti) of >8.7 cm post-cardioversion indicate maintenance of sinus rhythm. Therefore, echocardiography can help in making decisions about whether to pursue a rhythm or rate control strategy in these patients.

In a study of 209 patients, M. Balik et al. showed the significant role of echocardiography in predicting outcomes for patients with newly diagnosed supraventricular arrhythmia and septic shock. Among the patients in the study, 64.1% of patients experienced at least one recurrence of supraventricular arrhythmia after drug cardioversion. By using echocardiography 4 h after cardioversion, it was found that a LA-EF <38.4% and Avti ≤6.8 cm were predictive of a single arrhythmia recurrence. On the other hand, a sustained sinus rhythm was associated with a LA-EF >44% and Avti >8.65 cm, with these markers showing an inverse relationship with the number of recurrences. Factors such as the enlarged left atrial end-systolic diameter at the onset of arrhythmia and the increased systolic pulmonary artery pressure at 4 h were also considered predictors for multiple recurrences [32].

The observation of side effects of antiarrhythmic drugs, as well as the recurrence of arrhythmias, increasingly highlights the uncertainty between choosing rhythm control or frequency control [32,38]. Current evidence suggests better efficacy for heart rate control in combination with effective anticoagulation, but rhythm control should not be overlooked [32,39].

Both the double-blind prospective randomized controlled study [1] and the post-hoc analysis of 209 patients [32] demonstrated that propafenone achieved better and faster rhythm control compared to amiodarone in septic shock patients with a non-dilated left atrium.

Echocardiography is now the preferred imaging modality for investigating hemodynamically unstable patients in the ICU, and plays a crucial role in deciding on the appropriate antiarrhythmic medication [32,40,41,42]. The prognosis of patients with septic shock is closely linked to the diastolic function of the left ventricle, which can be assessed through echocardiographic evaluation of the left atrium. This comprehensive assessment contributes to the decision whether to control the rhythm and restore atrial systole, or to control the rate. Despite a high incidence of septic shock patients with NOAF, there is a lack of studies examining predictors of cardioversion in this population [32].

Expert data on risk factors for the occurrence of atrial fibrillation in ICU patients are limited. The need for catecholamines and positive inotropic medications, commonly used in patients with advanced septic shock, high disease severity index scores, and sepsis, are primary risk factors. Additionally, cardiovascular disease, electrolyte imbalances, elevated inflammatory markers, advanced age, hypoxia, and high central venous pressure are significant risk factors that should not be overlooked.

Despite atrial fibrillation not being a new diagnosis, it has been shown to be an independent risk factor for mortality in intensive care [43,44].

When choosing between controlling rhythm or frequency, the literature lacks clear data on which perspective is preferable. However, for patients with NOAF and a dilated left atrium, amiodarone is preferred [1,24]. Nonetheless, amiodarone does have significant side effects, as indicated by various studies, and the use of amiodarone in patients with septic shock should be considered only if propafenone is unavailable and the patient has no other associated cardiac pathologies (e.g., a dilated left atrium) [17].

For critically ill patients, including those with septic shock who do not respond to fluid resuscitation and require vasopressor support, the use of class 1C antiarrhythmic drugs like propafenone has not been thoroughly studied. Limited case reports suggest there may be adverse effects related to cardiotoxicity, but evidence is lacking. Comparisons between propafenone, amiodarone, and metoprolol in septic shock patients are scarce in the literature [17,22,25,27,28,29]. Studies examining heart rate control in critically ill NOAF patients have compared the effects of amiodarone with digitalis medications. A study published in 2022 found that amiodarone reduced heart rate to <110b/min in 2 h, faster than digitalis, with better maintenance of sinus rhythm in the first 24 h of treatment. However, patients receiving amiodarone experienced more bradycardia episodes (7.7% vs. 3.4%), and an increased need for noradrenaline (23.9% vs. 12%) [45]. The use of amiodarone as a first-line treatment for septic shock patients remains uncertain, especially considering the frequent need for noradrenaline in these cases. The 2022 study suggests that amiodarone may exacerbate the need for noradrenaline, potentially worsening patients’ outcomes. The optimal medication for heart rate control in these patients is still a topic of debate, with no clear recommendations provided in the 2020 guidelines for acute heart failure management by the European Society of Cardiology [40].

Various studies show the significant effects of both amiodarone and propafenone in restoring sinus rhythm, depending on the underlying cardiac pathology. For patients with a dilated left atrium, intravenous amiodarone has been found to provide better rhythm control until cardioversion occurs, but with a higher rate of recurrence in this patient category. Conversely, for patients without a dilated left atrium, oral propafenone has been shown to achieve cardioversion 3.7 h faster than amiodarone (6.4 h), with lower recurrence rates (52%) [27].

Regarding the dose of antiarrhythmic medications administered, particularly in the case of amiodarone, it is directly correlated with the need for continued antiarrhythmic treatment at home. The higher the dose of amiodarone given in the ICU, the higher the likelihood of requiring continued medication post-discharge. A study of 239 patients also revealed that the intravenous dose of amiodarone administered in the ICU is an independent factor associated with an extended duration of hospitalization [1,26,29].

Despite being a cornerstone in arrhythmia treatment, amiodarone was not the most commonly used drug for NOAF occurring in sepsis, as indicated by a 2016 study. Calcium channel blockers were ranked first in usage, with the group of patients receiving this medication also showing the highest mortality rate. The group receiving beta-blockers had the lowest mortality rate, with the amiodarone group ranking second [25].

For NOAF, one of the recommended management strategies is electrical cardioversion. After the procedure, antiarrhythmic medications are recommended to maintain sinus rhythm. However, there are limited data in the literature regarding antiarrhythmic recommendation practices.

A study published in 2024, conducted over a 5-year period (2015–2020) with a sample of 223 patients, showed that amiodarone was the main antiarrhythmic recommended after electrical cardioversion, with sotalol and flecainide (36.5% vs. 27.8% and 1.3%, respectively) in second place. Interestingly, for 35.2% of patients, no antiarrhythmic was recommended [46]. Amiodarone has become the most widely recommended antiarrhythmic over the last 40 years [47]. Amiodarone and sotalol have shown similar efficacy and for patients who did not receive antiarrhythmic therapy, their maintenance of sinus rhythm at 12 months after electrical cardioversion was comparable to that of the amiodarone and sotalol groups [48].

When considering the ability of amiodarone to restore sinus rhythm in patients with non-valvular atrial fibrillation, a meta-analysis of 13 randomized trials demonstrated the superiority of amiodarone over a placebo, but with reduced efficacy compared to class I antiarrhythmic medications. Whitin the first 8 h, amiodarone administration is less effective than class I medication in restoring sinus rhythm, but 24 h after administration, the efficacy of the two is similar [2]. Despite its antiarrhythmic efficacy, both intravenous and oral administration of amiodarone may lead to serious adverse effects.

A study published in 2022 reported the case of a 70-year-old patient who experienced refractory hypotension associated with renal failure and acute hepatitis after intravenous administration of amiodarone [49]. Amiodarone is known to have high hepatic, thyroid, renal, pulmonary, ocular, and integumentary toxicity. All these risks should be carefully considered before administering this antiarrhythmic drug [45,50].

The goal of pharmacotherapy is to reduce morbidity and to prevent complications [51].

Another major problem faced by intensive care clinicians is the increased risk of venous catheter-associated complications that patients in this category are exposed to, along with an unclear risk-benefit balance for anticoagulant administration. A retrospective cohort study based on administrative data found a significant increase in the risk of bleeding in septic patients with NOAF who were treated with parenteral anticoagulants [52].

## 5. Conclusions

In conclusion, summarizing several studies, it is shown that follow-up of patients who have undergone drug cardioversion contributes to the therapy of diastolic dysfunction and has a positive impact on mortality.

A frequent and potentially life-threatening complication is NOAF, occurring in one in seven patients with sepsis, with the rate increasing among patients with septic shock. Further studies are needed to clarify the best strategy for managing patients with septic shock and NOAF, considering both the acute pathology and the patient’s underlying disease. Until then, it is up to each physician to decide whether to control the rhythm or frequency by administering different classes of antiarrhythmic treatment. In addition to these data, for patients without other underlying cardiac pathologies, propafenone should be considered, or, if the situation permits, the use of electrical cardioversion rather than amiodarone.

## Figures and Tables

**Figure 1 medicina-60-01436-f001:**
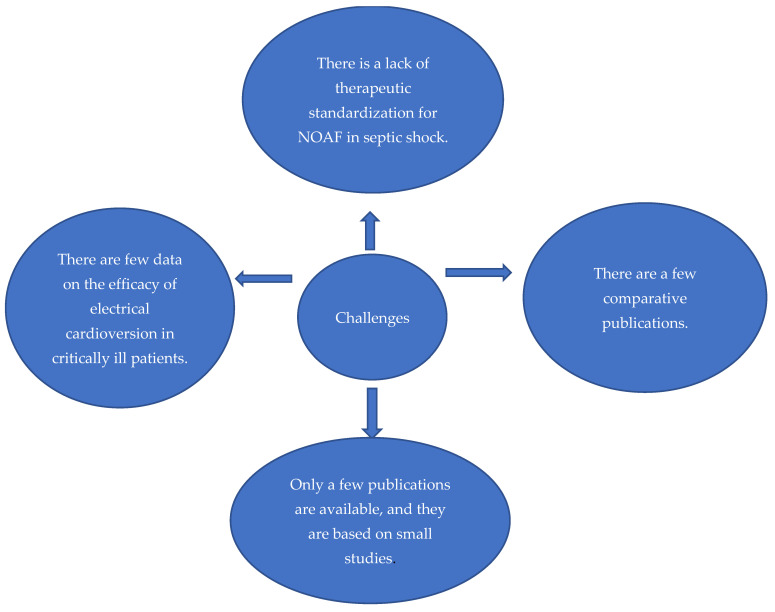
Challenges in restoring sinus rhythm or heart rate control in patients with septic shock and newly diagnosed atrial fibrillation.

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
