# Peer review of "Evaluation of Amiodarone Administration in Patients with New-Onset Atrial Fibrillation in Septic Shock"

_medicina, 2024, doi:10.3390/medicina60091436_

Round 1
Reviewer 1 Report
Comments and Suggestions for Authors
The text turns out to be very interesting.
However, it is worth looking into some aspects related to the Amiodarone molecule.
Amiodarone has a lower capacity than IC to pharmacologically cardiovert AF, however in the acute setting it appears to be more manageable and safe (being able to be administered to a wider audience of patients, including cardiac patients). This concept must be broadened into question and placed with reference .
Furthermore, a chapter should be made on the ability to maintain sinus rhythm after electrical cardioversion which represents a common setting in ICUs.
A reference must then be made to the toxicity of Amiodarone (low in acute administration)
Author Response
I .corrected

Reviewer 2 Report
Comments and Suggestions for Authors
General concept comments
The aim of the present review is to assess whether the administration of Amiodarone is the safest way to control heart rate, in septic shock patients.
The testability of the hypothesis, methodological inaccuracies, missing controls are not presented cause of it is review article.
The review topic is actually. Atrial arrhythmias (AA) are associated with hemodynamic compromise in septic shock and two to five times increased mortality among critically ill patients.
Electrical cardioversion is recommended in hemodynamically unstable patients and is more efficient in combination with an antiarrhythmic agent due to the high rates of an early relapse of AA.
Observed recurrence of arrhythmias and the side effects of the antiarrhythmics bring uncertainty whether to aim for rate control rather than for rhythm control therapy.
Balik M. et al. study showed a high cardioversion rate with faster achievement of sinus rhythm and fewer recurrences in the propafenone arm compared to amiodarone [M. Balik, M. Maly, T. Brozek, J. Rulisek, M. Porizka, R. Sachl, et al. Propafenone versus amiodarone for supraventricular arrhythmias in septic shock: a randomised controlled trial Intensive Care Med, 49 (11) (2023), pp. 1283-1292.].
P. Waldauf et al. study suggests that patients with non-dilated left atrium who achieved rhythm control with propafenone in the setting of septic shock had better short-term and long-term outcomes than those treated with amiodarone [P. Waldauf, M. Porizka, J. Horejsek, M. Otahal, E. Svobodova, I. Jurisinova, et al. The outcomes of patients with septic shock treated with propafenone compared to amiodarone for supraventricular arrhythmias are related to end-systolic left atrial volume. Eur Heart J Acute Cardiovasc Care (2024), 10.1093/ehjacc/zuae023.].
The manuscript is clear, relevant for the cardiovascular disease field and presented in a well-structured manner.
References are appropriate. 49 (100.0%) references are given, of which only 24 (48.9%) are less than 5 years old. An optional recommendation is to replace some outdated references with newer ones, preferably no more than 5 years old and correct publication date of some references.
Specific comments
On line 29 the authors probably meant not “…been reached. Thaditional…”, but “…been reached. Traditional…”.
On line 30 the authors probably meant not “…canal blockers, commonlu used…”, but “…canal blockers, commonly used…”.
On line 39 the authors probably meant not “…over 6days face significantl higher…”, but “…over 6 days face significantly higher…”. And what does it mean: “6 days face”?
On line 48 the authors probably meant not “…Those with septis who developed…”, but “…Those with sepsis who developed…”.
It is necessary to decipher the abbreviation “FIA” in line 57.
On line 79 the authors probably meant not “…were then compliled into electronic…”, but “…were then compiled into electronic…” or “…were then complied into electronic…”.
On line 91 the authors probably meant not “…papaers directly…”, but “…papers directly…”.
What does it mean “postsarche” on line 119?
The abbreviation “NOAF” is first encountered in the text on line 126, and the abbreviation's explanation is given only on line 238. The explanation should be moved to line 126.
On line 132 the authors probably meant not “…studies showted that…”, but “…studies showed that…”.
On line 188 the authors probably meant not “…patients witha dilated…”, but “…patients with dilated…”.
On line 208 the authors probably meant not “…However,r patients…”, but “…However, patients…”.
On line 209 the authors probably meant not “…with Propafenonehad less…”, but “…with Propafenone had less…”.
On line 317 instead of “…frequency control…” it is better to use “…heart rate control…”.
On line 327 the authors probably meant not “…is closely linkted to the…”, but “…is closely linked to the…”.
On line 336 the authors probably meant not “…venous pressure aresignificant risk factors…”, but “…venous pressure are significant risk factors…”.
On line 377 the authors probably meant not “…by a 2016 stydy…”, but “…by a 2016 study…”.
On line 379 the authors probably meant not “…also showring the highest…”, but “…also showing the highest…”.
On line 382, the abbreviation “VCА” is unnecessary, since it does not appear further in the text.
The text contains a lot of double and triple spaces, and punctuation marks are missing.
Author Response
I corrected

Reviewer 3 Report
Comments and Suggestions for Authors
Oprea et al have submitted a review article discussing role of Amiodarone in new onset atrial fibrillation. I have several about the manuscript that are listed below:
NOAF is used in title, and through early stages of manuscript without being clear what the acronym means. Please use full term in Title, and introduce the term in the Introduction section before using the acronym. Similarly, acronym FIA is used on Page 2, Line 57, but no explanation is provided as to what it stands for. It might be easily understood by cardiologists, but can be very problematic for readers that are unfamiliar with these acronyms.
Authors state that "Amiodarone is a relatively new medication". Relative to what? By what standards? It was introduced in the 1960s so that would make it more than 50 years old. That would not be "new" by current standards where new drugs are being introduced to the market every month.
In figure 1, the last line in Left sided bubble is cutoff and it is unclear what it says.
There are too many typographical errors in the manuscript, at times making it impossible to understand what the authors are trying to say. For example, in line 119, what does "postsarche" mean?
Section 2.1.1 simply regurgitates facts and findings from other studies, and although reporting published literature is a core aspect of a review article, the authors also need to add some intellectual input and their own synthesis of information into the manuscript.
In line 234-236, the authors state that the "mainstay of antiarrhythmic therapeutic is Amiodarone" and cite reference 33. This is a very bold and aggressive claim, however, this reference is simply a protocol for a RCT and not an RCT itself. Therefore it is absolutely inappropriate to use this reference to support such a bold and blanket claim.
My biggest concern with the paper is the entire premise that is setup by the authors that amiodarone is hemodynamically friendly and safer to use in septic patients with NOAF. They themselves state in line 61-62 that Amiodarone can be used without concerns for worsening cardiac function or hypotension. However, they then report in line 358 that there is a higher need for Noradrenaline in patients receiving Amdiodarone, thereby leading to worse outcomes. This kind of contradictory messaging by the authors makes the manuscript almost incoherent, and deprives it of any scientific value.
Comments on the Quality of English LanguageA lot of typographical errors noted throughout the manuscript.
Author Response
I corrected

Round 2
Reviewer 1 Report
Comments and Suggestions for Authors
"For chronic atrial fibrilation(AF) , one of the recommended management strategies Is 383 electrical cardioversion"
chronic AF doesn't exist! Pleas chek.
Permanent AF is the right definition and it is unsuitable for Cardioversion. Please revise
Author Response
This is my reply.

Reviewer 3 Report
Comments and Suggestions for Authors
The authors Oprea et al have re-submitted the paper, but it appears that they have not changed the premise of the paper. They have not given a pont-by-point response to the the concerns that I raised. It would seem that they have just removed some of the sentences that I was concerned about, but I cannot be sure.
In lines 281-285 they list the known adverse effects of Amiodarone. But they do not report one of the most relevant and commonest side effct: hemodynamic compromise. Below reference shows that Amiodarone is a negative inotrope and decreases cardiac output, neither of which are desired issue in sepsis and spetic shock. https://www.ncbi.nlm.nih.gov/pmc/articles/PMC6980379/#R17
Again, in line 344 they state that AMiodarone is preferred due to lower cardiodepressant effect. However, this is not supported by appropriate references that support its use in septic patients.
They cite reference 18 to state: "its effectiveness in patients with septic shock is still uncertain". However, if you review the actual paper cited, in the conclusion section of the abstract it is clearly stated that Propafenone was superior to Amiodarone. I am not sure how they are calling it "uncertain".
In summary, the authors seem to believe and want to get a message across that Amiodarone is useful for NOAF in sepsis, but the available data does not support that. And writing a review article will not accomplich their goal. Perhaps they should perform their own study that explores Amio for NOAF in sepsis, and publish their results.
Comments on the Quality of English LanguageSee above.
Author Response
This is my reply.
Thank you for your collaborations.

Round 3
Reviewer 3 Report
Comments and Suggestions for Authors
The authors have made significant changes and now included text that is less forceful towards Amiodarone, and has more of a "suggestive" tone to it. I agree to the changes.
Comments on the Quality of English LanguageAcceptable.